# Models of Congenital Adrenal Hyperplasia for Gene Therapies Testing

**DOI:** 10.3390/ijms24065365

**Published:** 2023-03-10

**Authors:** Olga Glazova, Asya Bastrich, Andrei Deviatkin, Nikita Onyanov, Samira Kaziakhmedova, Liudmila Shevkova, Nawar Sakr, Daria Petrova, Maria V. Vorontsova, Pavel Volchkov

**Affiliations:** 1Gene Editing Laboratory, Endocrinology Research Centre, 117292 Moscow, Russia; 2Genome Engineering Laboratory, Moscow Institute of Physics and Technology, 141700 Dolgoprudniy, Russia

**Keywords:** adrenal cortex, cell differentiation, steroidogenic factor 1, CAH, organoids, murine models, single cell RNA sequencing

## Abstract

The adrenal glands are important endocrine organs that play a major role in the stress response. Some adrenal glands abnormalities are treated with hormone replacement therapy, which does not address physiological requirements. Modern technologies make it possible to develop gene therapy drugs that can completely cure diseases caused by mutations in specific genes. Congenital adrenal hyperplasia (CAH) is an example of such a potentially treatable monogenic disease. CAH is an autosomal recessive inherited disease with an overall incidence of 1:9500–1:20,000 newborns. To date, there are several promising drugs for CAH gene therapy. At the same time, it remains unclear how new approaches can be tested, as there are no models for this disease. The present review focuses on modern models for inherited adrenal gland insufficiency and their detailed characterization. In addition, the advantages and disadvantages of various pathological models are discussed, and ways of further development are suggested.

## 1. Introduction

The adrenal cortex is the most important site for the synthesis of steroid hormones. It is histologically and functionally divided into layers surrounding the central medulla. The zona glomerulosa (ZG), which produces mineralocorticoids (mainly aldosterone), is located directly under the capsule that covers the entire gland. The zona fasciculata (ZF), which synthesizes glucocorticoids (cortisol in humans and corticosterone in rodents), is located behind ZG, as is the zona reticularis (ZR), which produces adrenal androgens (in humans and some other mammals). Mineralocorticoid production is controlled by the renin-angiotensin-aldosterone system, while glucocorticoids are under the control of the hypothalamic-pituitary-adrenal system. Glucocorticoids regulate glucose metabolism, inflammation, immune responses, muscular and skeletal mass, as well as cognition, well-being, and memory. Mineralocorticoids control extracellular fluid volume and sodium homeostasis and therefore have an important effect on blood pressure [1].

The adrenal cortex is an organ that in rodents (and presumably in humans) is capable of self-renewing throughout life, replacing senescent cells, and maintaining or expanding functional areas to meet physiological needs for steroids, or in response to external pharmacological stimuli [2,3,4]. Adrenal insufficiency occurs when the adrenal cortex does not produce sufficient amounts of glucocorticoids, with or without simultaneous mineralocorticoid and androgen deficiencies, for example in congenital adrenal hyperplasia (CAH) [5,6]. CAH is an autosomal recessive disease, with an overall incidence of 1:9500–1:20,000 newborns, caused by mutations in the genes encoding enzymes for various stages of steroid biosynthesis such as *StAR*, *CYP11A1*, *CYP11B1*, *HSD3B2*, *CYP17A1* (Figure 1) [6,7]. To date, more than 30 other causes of primary adrenal insufficiency have been registered, including genetic disorders whose mechanisms are gradually being elucidated [8].

### Adrenal Insufficiency Treatment Approaches

Currently, the only available treatment for patients with adrenal insufficiency is hormone replacement therapy, which in most cases must be lifelong. The administration of exogenous glucocorticoids compensates for the lack of endogenous cortisol and acts through negative feedback from the hypothalamus and pituitary gland by suppressing the secretion of corticotropin-releasing hormone (CRH) and ACTH. The disadvantage of traditional replacement therapy is that choosing the correct dosage of medication is extremely difficult and does not reproduce the physiological rhythm of hormone release and an incorrect dosage can lead to negative consequences with prolonged treatment [9]. Despite the fact that recent efforts of the scientific and medical communities have been focused on the new glucocorticoid delivery systems that could better reproduce their physiological levels, or on alternative therapeutic approaches that do not involve glucocorticoids, the correction of the symptoms of adrenal insufficiency remains an urgent issue [10].

Gene therapy is an experimental technique in which DNA or RNA is introduced into the human body to alter gene expression in order to treat or prevent a range of diseases. Gene therapy approaches were first proposed almost 40 years ago, and the first meaningful studies were conducted about 30 years ago [11]. The new approaches offer hope in cases where diseases are considered imperfectly treatable with conventional medicines. A number of hereditary diseases can be distinguished by whose etiology is known; the pathology occurs as a result of a disturbance in the work of one gene. If a functional copy of the gene is supplied to the human body with the help of a gene therapy drug, the development of the disease can be slowed down, and in some cases, even a significant improvement can be achieved. CAH is an example of such a monogenic disorder, caused by mutations in the *CYP21A2* gene. *CYP21A2* is an enzyme that is necessary for the synthesis of aldosterone and cortisol (Figure 1).

There is a gene therapy approach for the treatment of CAH that is already in phase ½ of clinical trials. The strategy developed by the company Adrenas is based on the intravenous administration of AAV5 carrying the wild-type coding sequence of the *CYP21A2* gene (named BBP-631). Preclinical studies with BBP-631 in the *Cyp21-/-* mouse model and non-human primates showed strong rescue of steroidogenesis and the ability of the virus to transduce adrenal cells [12].

The use of cell and gene therapy strategies to create or support functional adrenal tissue is a promising alternative to existing methods of treating primary adrenal insufficiency, regardless of disease etiology, as this approach allows the organ to function more physiologically. However, these approaches require species-specific disease models, as different animals differ physiologically (see below).

Although the development of cell therapy strategies in the adrenal region is still in its infancy, it is important to recognize that some significant results have been achieved to date. The following sections discuss current approaches and advances, including the use of adrenal cells for transplantation, the use of functional steroidogenic cells derived from stem cells or reprogrammed somatic cells, and the creation of three-dimensional models of adrenal organoids (Figure 2).

## 2. Congenital Adrenal Hyperplasia Models

### 2.1. Monolayer Cells

There are a few approaches to obtaining steroidogenic tissue that have been developed (Figure 2A). Most of them are based on the differentiation of mesenchymal and embryonic stem cells (MSC and ESC, respectively) With regard to human embryonic stem cells or induced pluripotent stem cells, two approaches have been used to induce steroid-producing cells: by embrionic bodies formation and by using the multistep mesodermal differentiation method [13]. Furthermore, direct differentiation of ESC resulted in increased cell death due to overexpression of steroidogenic factor 1 (SF1) [14]. For this reason, MSCs began to be used as a cell source for mimicking steroidogenic tissue.

MSCs are multipotent somatic stem cells that originate in the mesoderm, as is the case with steroidogenic organs. Although originally identified in bone marrow [15], similar populations have been reported in many other tissues such as adipose tissue, umbilical cord blood (UCB), and the placenta. MSCs may be a source of connective tissue lineages in vivo. In other words, MSCs are present in most organs of the human organism, with the potential for application in regenerative medicine. Both murine and human MSCs have been shown to be capable of producing steroid hormones when forced to express SF1, in contrast to several other stable cell lines. Nevertheless, there are some species-specific molecular and functional differences between mouse and human cell sources. In contrast to murine MSCs, human MSCs demonstrated strong expression of the *CYP21A2* gene, causing a difference in the types of steroids produced by murine MSCs and human MSCs. Glucocorticoids were the major steroids produced by transformed human MSCs, while testosterone was the major product from the transformed murine MSCs. Furthermore, murine MSCs express the adipose tissue type of adrenocorticotropic hormone receptor (with low ACTH affinity), whereas human MSCs did not express a special isoform of the receptor that could respond to ACTH [16].

Despite the evidence that human MSC from different sources can be induced to be steroidogenic, differences in the differentiation process of the initial tissues were observed [17,18]. For example, in contrast to bone marrow cells, adipose tissue-derived mesenchymal cells were much more likely to produce an adrenal steroid, such as corticosterone, than a gonadal steroid, such as testosterone [16,19]. The human umbilical cord represents an innovative uncontroversial source of mesenchymal stem cells (UCB-MSCs). UCB-MSCs can be easily obtained because their collection, unlike the isolation of bone marrow, does not require invasive procedures. Stable expression of SF1 in conjunction with cAMP treatment in human UCB-MSCs has also been shown to be capable of initiating a steroidogenic program. At the same time, a strong similarity with granulosa luteal cells and especially with ovarian granulosa cells is shown. Therefore, this approach cannot be used for modeling specific adrenal tissue [20]. Both human UCB-MSCs and human bone marrow (BM) MSCs had typical MSC characteristics, according to the criteria described by the International Society for Cellular Therapy. Differentiated human UCB MSCs were shown to have significantly higher viability than bone marrow MSCs after infection with an adenovirus containing SF1 and higher expression of all steroidogenic mRNAs tested for. Furthermore, differentiated human UCB-MSCs secreted more testosterone and cortisol compared to differentiated human bone marrow MSCs [21]. It should be noted, however, that there are significant differences in the general gene expression pattern of MSCs from bone marrow, adipose tissue, and UCB, and that many genes in MSC from different ontogenetic sources or under different culture conditions are differentially expressed [22]. This means that MSCs from different tissues should not be considered as a single cell type.

As mentioned above, for the most part, protocols for differentiating stem cells into the steroidogenic lineage are based on the forced expression of steroidogenic factor 1 (SF1). This gene, encoded by nuclear receptor subfamily 5, group A, member 1 (*NR5A1*), regulates steroidogenic pathways, including genes encoding cytochrome P450 enzymes such as *CYP11A1* [23], *CYP17A1* [24,25], *CYP21A2* [26], *CYP11B1* [27], 3-hydroxysteroid dehydrogenase (*3-HSD*) [28], and other proteins important in steroidogenesis. SF1 also regulates genes involved in sex determination and the development of reproductive tissues including *DAX1*, *AMH*, *SOX9*, *SRY*, and *INHA* [28,29,30,31,32]. At the same time, differentiation into the steroidogenic lineage may be induced by retroviral expression of *LRH-1* [33,34]. Liver receptor homolog-1 (LRH-1) belongs to the same protein subfamily as SF1 and has a very similar structure and function. Thereby, the currently available protocols for obtaining steroidogenic cells still require the exogenous expression of SF1 or SF1 homologues, which makes the protocol technically more complicated and unnatural. This protocol should therefore be overcome in the future. In addition to the requirement for SF1 overexpression, 8-Bromoadenosine 3′,5′-cyclic monophosphate has been shown to be an important component of the induction medium that regulates steroidogenic pathways during cAMP stimulation.

The same differentiation method was used for CAH patient-specific cells (urine derived MCS or iPSCs from peripheral lymphocytes, that was firstly directed to mesodermal lineage) with different mutations in the *CYP21A2*, *CYP17A1* and *CYP11B1* genes. The steroids produced by the differentiated cells carrying mutations had a pathogenic profile similar to those demonstrated during disease progression (high levels of progesterone and 11-deoxycorticosterone and low levels of cortisol). Treatment with lenti- or adeno-associated viruses carrying the normal corresponding gene copy restored normal steroidogenesis [35,36] (Table 1).

### 2.2. Three Dimensional Structures

Other hopeful approaches to create the models of the normal and pathogenic adrenal tissue include three-dimensional structures such as spheroids, organoids, or bioartificial organs that represent and simulate the microenvironment and structure of real organs and tissues [44] (Figure 2B). These structures have been shown to produce more steroids and relevant enzymes compared to monolayer cell cultures [45]. A spheroid culture is a 3D multicellular culture, representing a similar physicochemical environment, the cell to cell and cell to matrix interactions in vitro. Such a model takes on the disadvantages of monolayer cell cultures [46]. Adrenocortical spheroids from the adrenals cortex tissues of 2–5 month old mice cultured with insulin during a one week differentiation period showed increased expression of *Cyp11b1*, *Cyp11a1*, and *Nr5a1*; they produced 11-deoxycorticosterone and corticosterone. Further differentiation into ZG cells was evidenced by significantly increased aldosterone production on day 9 of incubation [47].

An organoid is a multicellular structure that represents and simulates the microenvironment and structure of real organs or tissues, consisting of the different cell types [44]. Organoids of the adrenal gland were obtained from aborted first-trimester fetuses, cultured on plates for 7 to 10 days and then tended to form three-dimensional detached structures. These organoids consisted of two zones: small highly proliferating cells positive for neuroendocrine progenitor (Nestin) and differentiated cortex protein markers (CHGA, and SF1) and the inner zone positive for *CYP17A1* with some groups of tyrosine hydroxylase positive cells. Interestingly, the presence of flattened cells at the periphery of adrenal organoids was discovered, some of which contained lipid droplets, suggesting the organization of a rudimentary adrenal capsule primordium [48].

It is well-known that matrix signaling is highly important for cell maintenance and differentiation. One of the most popular methods for producing the three-dimensional organoid structures is the use of a decellularized adrenal extracellular matrix (ECM). Allen et al. used porcine adrenal glands as a biologic scaffold for human fetal (between 12 and 16 weeks gestation) adrenal gland cells. The simple mix of the scaffold and cells yielded the attachment of most cells onto the surface of the scaffold, with rare cell infiltration events. It is interesting that the decellularization process removed the laminin molecules from the ECM instead of collagen IV and fibronectin, but manual addition of laminin (known to be an important component of the stem and differentiated cells niche) had no effect on the efficiency of the cellurization or cortisol production of the obtained artificial organ [49]. Five years later, another research group tested bioartificial adrenals prepared from bovine adrenal cells and alginate (bovine adrenocortical cells (BACs) in alginate–enBAC) in vivo on adrenalectomized rats [50]. Notably, the protocol called for separate extraction of the cortex and chromaffin cells and their simultaneous encapsulation. The approach had no influence on cortisol release, but significantly increased BAC viability. enBACs were able to produce cortisol until at least day 70 of cultivation and did not lose their ability to respond to ACTH, in contrast to a total loss of the ACTH-stimulated secretion of cortisol by regular BAC cultures after 11 days. These results are in agreement with the stem cell presence in BAC cultures that was found by the same group [51]. In an immunodeficient rat model, enBAC implanted animals showed approximately two times normal weight gain and cortisol level that was stable in the enBAC group and decreased after 11 days in the control (BAC cells without alginate) group. Moreover, heavy vascularization of the enBAC implants was observed. Immunocompetent adrenalectomized rats also showed high levels of enBAC implantation treatment efficiency. It is notable that the bovine cells were efficiently protected by alginate from the immune system that was shown by the 90% viability of the cells at day 21 of the experiment and a stable ACTH-induced response. In contrast, using oxygenated immune-isolating devices for BAC in vivo delivery yielded less clinical efficiency [50]. Stem cells presence in the BAC culture were shown and different progenitor cell populations maintained at enBAC for at least 114 days.

It should be noted that SF1-driven differentiated cells embedded in alginate showed completely opposite behavior in comparison to those obtained from organs [35]. Human induced steroidogenic cells underwent necrosis over one week after being coated in alginate, while BAC are stable for over 100 days.

Modern approaches in molecular biology give scientists the opportunity to work with human stem cells and differentiate them into almost any somatic cell. Developing human adrenal glands from stem cells using various differentiation methodologies is the future of adrenal disease modeling and insufficiency treatment.

The aforementioned models could be used to model normal adrenal tissue. At the same time, there are several approaches to inhibit steroidogenesis in hormonal hypersecretion [52]. It may be possible to apply some of these inhibitors to the normal tissue models to create a CAH model, which to the best of our knowledge has not yet been done.

### 2.3. Animal Models

As long as 3D human organ structures are not perfectly developed, there are several in vivo models of CAH that exist and are usable (Figure 2C).

The mechanical model of adrenal insufficiency used in the past to test different therapeutic approaches is the adrenalectomized animals [53]. This type of model is characterized by decreased glucocorticoids and mineralocorticoid production, body weight loss, and short survival term (about one week) [45,54]. Historically, it has been used to study the function of adrenal hormones and their influence on various processes and pathologies [55,56,57]. A study on CAH cell therapy used adrenalectomized mice and rats and showed that lifespan increased after transplantation of SF1-forced differentiated cells [45].

The first genetically modified model is the C57Bl/10SnSlc-H-2aw18 mouse strain, which carry the *Cyp21a2-p-Cyp21a1* hybrid gene with different mutations that completely switch off 21-hydroxylase activity. The mice have a pronounced CAH phenotype, which is manifested by a loss of corticosterone and aldosterone production, weight loss, and increased level of progesterone. According to the NCBI blast analysis, there is a 78% nucleotide and 73% aminoacid sequence difference between the mouse and human *Cyp21a2* genes. For this reason, it remains unclear whether it is correct to treat mice with the human Cyp21 protein or gene. Restoration of 21-OH activity by the human gene in Cyp21A2-deficient mice has been shown to restore the normal phenotype [37]. In order to fully clarify the question of the *CYP21A2* gene functional compatibility between humans and mice, humanized mice were created [38]. No adverse effects of the human gene on mouse development, health, and fertility were found, suggesting that the mice may be a suitable model for *Cyp21A2* gene based studies.

Another CAH mouse model was created by using CRISPR/Cas9 mediated genome editing that disrupted the 4th exon in mice with the *Cyp11b1* gene. These mice represent an increased 11-deoxycorticosterone and progesterone to corticosterone ratio that can be rescued by intraadrenal AA9-Cyp11b1 induction [36].

Two mouse models for a particular type of congenital adrenal hyperplasia—lipoid CAH—were also obtained and studied. The causes of this type of CAH are mutations in the genes whose protein products are at the beginning of the steroidogenic pathway–*StAR* and *CYP11A1* (Figure 1). These disorders affect not only the adrenal glands but also other steroidogenic organs such asthe gonads. Mice in which the StAR gene has been knocked out show the severe phenotype of adrenal insufficiency, which is due to the absence of normal steroidogenesis and abnormal lipid deposition [39]. A highly similar phenotype can also be observed in mice with *Cyp11a1* mutations [40].

Although the mouse CAH model was implemented, it does not fully reflect the pathological processes in humans due to species-specific differences in steroid production between mice and humans. The adrenal glands of mice lack 17-hydroxylase activity, so there is no androgen and cortisol synthesis in the same way as in humans as well as virilization of the external genitalia. Rabbits have more in common with human adrenal steroidogenesis [58] but only one model has been described. *CYP11A1*-affected rabbits show a lipoid CAH phenotype that correlated with that of human patients [41]. Thus, mammals (besides mice) may be the models for CAH, that share the same molecular processes and adrenal physiology as humans [59].

Several other vertebrate animal models of CAH have been created, showing that it is of interest not only for the study of pathology but also for the study of evolutionary processes. Firstly, a 21-OH deficiency modeling was carried out on animals from different taxons. For example, the deletion of 11 base pairs from the *cyp21a2* gene of *Xenopus tropicalis* generated with the CRISPR/Cas9 system resulted in a delay in the embryonic development of the tadpoles, but the metamorphosis process still went through [42,60,61]. The same effect of the *cyp21a2* mutation—glucocorticoid deficiency, disruption of the hypothalamic–pituitary–adrenal axis negative feedback and tissue hyperplasia—was found in the zebrafish knock-out model [43,62]. As in humans, cortisol is the main glucocorticoid produced by the adrenal gland, an analogue of the adrenal glands in zebrafish, so the fish model may be preferable to the mouse model in some studies.

## 3. Transcriptomics of the Adrenal Gland for Improvement Differentiation Strategies

As the adrenal gland is a complex organ, embryonic establishing and homeostasis of the tissue in adulthood is complex and not completely understood.

As mentioned earlier, the adrenal glands consist of three functionally distinct zones:ZG, which synthesize mineralocorticoids; ZF, which produces glucocorticoids; and ZR, which secretes androgens. This means that the transcriptional profiles of cells from the different zones that determine which proteins should be synthesized should be different. Indeed, Rege et al. found several genes whose differential expression most likely determines the functional differences between ZF and ZR [63]. To create a model of CAH, it is not necessary to develop a complete analogue of the adrenal glands. It might be sufficient to use an analogue of ZF cells. So far, however, it is still unclear how this could be produced. At the same time, it would be interesting to understand how the transcriptome profiles of the cells change during SF1-induced differentiation and which part of the adrenal tissue these cells most resemble. Measuring the expression levels of target genes for different differentiation methods in comparison to similar indicators of ZF could provide information on which differentiation method is optimal for CAH modeling. It should be noted that bulk RNA sequencing methods are not suitable to address this challenge. The adrenal gland contains a large number of different cell types with a unique transcriptional profile. Bulk sequencing, on the other hand, shows an average transcriptional profile for these cell types. The overall expression is therefore skewed and not associated with a specific cell type.

These aspects require careful analysis of constituent cell populations to develop a better in vitro differentiation protocol. In order to track the developmental trajectories of individual cell clones, it is important to identify rare cell populations representing complex systems in addition to the main populations. This, and establishing the relationship of regulation between genes, can be done through single-cell RNA sequencing, scRNA-seq [64]. scRNA-seq experiments allow the study of important intercellular variability and provide a clearer understanding of the influence of gene expression under more or less favorable conditions. Despite the fact that there are pioneering studies in which healthy adrenal glands have been examined on single samples using scRNA-seq of healthy adult adrenal glands [65], to our knowledge a detailed description of this tissue on a large sample dataset is not yet complete.

By identifying rare cell types using techniques such as scRNA-seq [66], it is possible to better understand differentiation paving the way for distinguishing between normal and pathological conditions at the cellular level. For example, gene expression markers for the differentiating ability of the iPSC lineage were identified in a study of molecular genetic variability at an early stage of human development [67]. A deeper understanding of the differentiation and development of embryos requires extensive data analysis, which this method provides by simultaneously processing a large amount of data on an object. In previous studies [68], many transcription factors and surface molecules were identified in the offspring of type 2 neuroblasts by targeted scRNA-seq sequencing of the Drosophila larvae brain third instar zone, which adds value to this study.

It is well known that the optimization of a differentiation protocol is a very laborious and time-consuming process. However, it is not possible to comprehensively investigate all signaling pathways. By labeling cells with individual “RNA” barcodes, new possibilities are opening up for studying the effects of activation of signaling pathways of different germ layers on gene expression. The data obtained were applied in the direction of ESC differentiation into a chord-like population [64]. The decisions of cell fate are driven by molecular mechanisms, the understanding of which can be improved by a combination of scRNA-seq analysis and genetic approaches. Using this approach [69], *KLF8* was identified as responsible for the transition from mesendoderm to endodermal progenitor cells. The development and differentiation of the embryonic heart in regenerative medicine is now being studied in more detail by identifying a cell type that expresses the key factor ETS1 [70]. These examples show that the application of new technologies can reveal new ways to overcome unsolved problems. Current studies based on adrenal scRNA-seq focus primarily on the medulla and associated pathologies, but some of the research also includes the adrenal cortex. The analysis of the adult adrenal gland demonstrated that well-known cell types persist inside the organ, such as adrenal fasciculate, glomerulose, capsule cells, as well as immune, endothelial, stromal cells, and others. Defined cell clusters exhibited high correlation across humans and mice, excluding the chromaffin population, which demonstrated a difference between these two species, namely one chromaffin cell cluster in humans and two in mice [65,71,72]. In the study of chronic stress, the scRNA-seq approach promoted the findings that the hypertrophy of the zona fasciculata cells is responsible for organ growth; there is a new group of cells with increased expression of *Abcb1b* that respond to stress adaptation [72].

The analysis of embryonic adrenal gland development at different time points is consistent with the current conception that the differentiation trajectory of the adrenal-gonadal-primordium (AGP) cells lies through the fetal stage and remain as fetal or can proceed to terminally differentiate. It was noted that the groups of fetal and definitive zone markers can be detected at low levels, even at the AGP stage of development [73]. In studies of the differentiation of the medulla, Schwann cell precursors have been found to generate chromaffin cells via an intermediate type of progenitor cells known as bridge cells, which are characterized by a specific transcriptional program [74].

Therefore, single cell transcriptome sequencing methods form the basis of projects describing structure and development of the mammalian organs [75,76]. Resolving molecular processes at the single cell level will improve understanding of such difficult and complex processes as embryonic development, adult stem cells differentiation and maintenance, and various pathologies. This knowledge will help researchers develop new approaches to therapies and model creation. At the same time, to our knowledge, no scRNA of the CAH adrenal gland has yet been performed. This means that the information about the cellular composition of the affected organ is not known. A comparison of the results of scRNA sequencing for different CAH models and the real pathology of the adrenal gland could ensure the choice of the most appropriate disease model.

## 4. Challenges in the Artificial Cellular Differentiation to the Adrenal Gland

It is clear that the development of therapies for adrenal pathologies based on cellular technologies is an urgent scientific and technological task. At this stage of development, the fundamental possibility of obtaining cell lines with some adrenal characteristics, as well as volumetric organ-like structures (organoids, spheroids, and artificial glands), was shown. However, there are still many gaps both in the developed protocols and in understanding the molecular basis of the ongoing processes.

For example, attempts to obtain steroidogenic cells in culture from various pluripotent cells have demonstrated that differentiation must consist of several stages, since it is difficult to change open ESC chromatin in one step to the expression of genes of the terminally differentiated state. Moreover, aggressive induction of SF1 in ESCs has been shown to be toxic [21]. Therefore, it is necessary to first partially commit the cells into the cells of the desired germ layer, or use MSC immediately, such as those isolated from urine. The cells could be directed to the state of terminal differentiation only after this stage.

At the same time, it is important to understand that terminally differentiated cells proliferate poorly, or even not at all. Although they remain viable for a long time in culture, it is not clear how this will work in vivo. The most obvious direction for further development is to focus on obtaining a culture of progenitors capable of proliferating and differentiating in vivo and in vitro in response to external signals.

The scientific community has yet to gain a definitive understanding of the sources and pathways of differentiation of adrenal progenitors into cells that produce different types of hormones. For example, when differentiation is mediated by exogenous expression of SF1, cortisol is primarily measured, demonstrating differentiation into the zona fasciculata, which contradicts the current ideas of the differentiation pathway going through the zona glomerulosa in general. In some studies [17], no aldosterone was found, indicating direct differentiation into zona fasciculata cells. Perhaps this is due to the active cAMP induction that is used in all protocols and stimulates zona fasciculata specific PKA pathway instead of WNT signaling, which is more active in the area of glomerulosis. scRNA should help to clarify the differentiation processes in detail and reveal the factors that regulate it.

In general, forcing exogenous expression of SF1 to obtain adrenal steroidogenic cells raises many questions. Different authors have used various delivery methods for this factor, including lentiviral transduction. In one study, using adeno-associated virus as a vector endogenous gene was shown not to be triggered [16], and in other studies, endogenous expression was not tested at all [35,77]. It is important to remember that SF1 demonstrates waves of expression in embryonic development. The difference in the control of SF1 expression in the fetal and definitive cortex hints that the regulation of this gene is complex and multifaceted. Apparently, in order to obtain a more physiological line of adrenal steroidogenic cells, it is necessary to induce the expression of endogenous *SF1*. Such attempts have been made by Ruiz through lentiviral expression of transcription factors thought to be regulators of SF1 [35]. Steroidogenic cells have not been obtained, but this method is worth further exploration. It is possible that differentiation of progenitors did not occur because the transduced cells were cultivated in a medium without the required inducers (cAMP). Given the preexisting expression of these transcription factors in stem cells, this may indicate their non-specificity in relation to the induction of SF1 and, therefore, other transcription factors should be selected (a combination of master regulators). It is likely that currently available single cell transcriptome sequencing data will help more accurately determine the master regulators of SF1 transcription at different stages of adrenal development and thus allow for finer control of the processes of differentiation in vitro.

Data from single cell transcriptome analysis can help to characterize more precisely the expression profile of progenitor cells in different tissues, making it possible to control the correctness of its implementation at different stages of in vitro differentiation. For example, the markers for hematopoietic stem cells are now well known and defined, and their presence in other stem cell populations is an important exclusion factor. Markers of progenitors of other cell types (e.g., gonadal and adrenal) are still poorly described, and this information is essential for cell type determination.

## 5. Conclusions

In the absence of appropriate CAH models, it is difficult to evaluate the performance of different drug prototypes. Although CAH models are under development (Table 1), it is currently hardly possible to assess which of the models is most plausible. At the same time, there are several ways to create such models, which were the subject of the current review. In conclusion, it should be noted that by themselves, the developed models could also become therapeutic products that are especially applicable to cellular technologies. Many protocols for obtaining adrenal-like structures are already being developed based on donor cells (iPSCs, MSC), although using primary cells for bioengineered products balances the advantages and disadvantages. Despite the patient-specific and more relevant human tissue and disease condition features of that model, there are still unpredictable challenges in the cultivation process, limited growth potential, and different cell behavior between donors. An alternative way is to use universal donor cells obtained from donors who are immunologically compatible with numerous recipients or genetically modified cells for the same reason. In any case, the development of technologies for obtaining specific organ structures is a perspective and promising task of biotechnology.

## Figures and Tables

**Figure 1 ijms-24-05365-f001:**
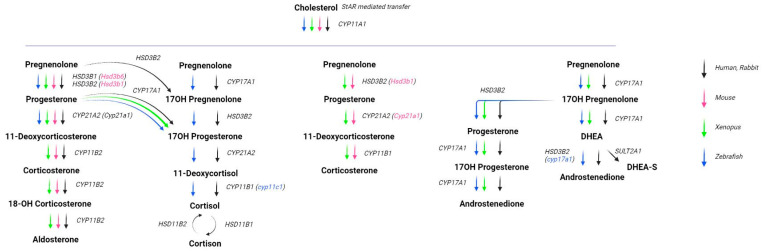
The known adrenal gland steroids synthesis pathways for the discussed models. Steroid precursors and products are denoted in bold. The genes and which protein products are responsible for the corresponding synthesis step, are denoted in italics. The names of the genes are mentioned in human mode, but if there is orthologous gene with different name, it is denoted in parentheses by color of the corresponding species. Created with BioRender.com.

**Figure 2 ijms-24-05365-f002:**
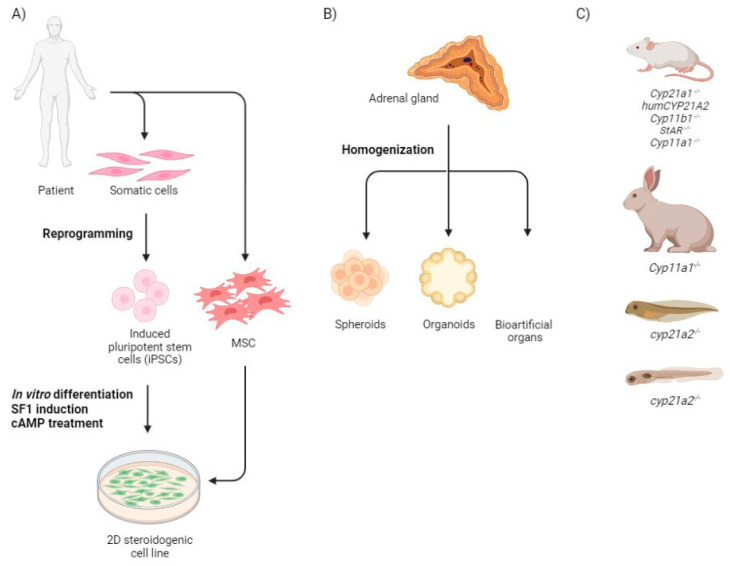
The main types of the CAH models. (**A**) Monolayer patient-specific differentiated cell cultures; (**B**) 3D structures from homogenized organ; (**C**) Genome modified animals (top down): five different mouse models with modified steroidogenic genes, rabbit *Cyp11a1* knock-out model, *Xenopus tropicalis* tadpole and zebrafish larva *cyp21a2* deficiency models. MSC–mesenchymal stromal cells: SF1–steroidogenic factor 1; dbcAMP–dibutyryl cyclic adenosine monophosphate; ECM–extracellular matrix; KO–knock-out. Created with BioRender.com.

**Table 1 ijms-24-05365-t001:** Developed CAH models.

Authors	Object	Mutated Genes	Mutations	Treatment	DOI
Ruiz-Babot et al., 2018 [35]	Steroidogenic cells differentiated from human urine-derived stem cells (USCs)	*CYP21A2*	c.515T > A, p.(Ile172Asn)	Lentiviruses	10.1016/j.celrep.2018.01.003
			c.955C > T, p.(Gln319Stop)		
		*STAR*	c.666delC, p.(Thr223Leufs × 98)		
		*HSD3B*	NA		
		*CYP11A1*	c.940G > A, p.(Glu314Lys)		
Naiki et al., 2022 [36]	Steroidogenic cells differentiated from human iPSc	*CYP21A2*	p.I172N/p.I172N	AAV2	10.1089/hum.2022.005
			p.R356W/IVS2–13A/C>G	AAV2	
			N.A.	AAV2	
			IVS2–13A/C > GR483delInt, CGG>CC	AAV2	
		*CYP17A1*	DF54/Y329KfsX418	AAV2	
		*CYP11B1*	p.W116X/p.W116X	AAV9	
Markmann et al., 2017 [37]	mice C57Bl/10SnSlc-H-2aw18	*Cyp21a1*	*Cyp21a1-Cyp21a2ps* chimera	AAVrh10	10.1089/hum.2017.203
Naiki et al. et al., 2022 [36]	mice C57BL/6-DBA/2/Cyp11b1-/-	*Cyp11b1*	The 4th exons disruped	AAV9	10.1089/hum.2022.005
Schubert et al., 2022 [38]	C57Bl/6NCrl-Cyp21a1*tg(CYP21A2)Koe*	*Cyp21a2*	replacing the 2620-bp *mCyp21a1*with the 2713-bp *hCYP21A2*	–	10.1210/jendso/bvac062
Hasegawa et al., 2000 [39]	mice C57BL/6/ *StAR*-/-	*StAR*	disrupted the *StAR* gene by deleting part of exon 2 and all of exon 3	–	10.1073/pnas.94.21.11540
Hu et al., 2002 [40]	mice C57BL/6/*Cyp11a1*-/-	*Cyp11a1*	neo gene was inserted within exon 1	–	10.1210/me.2002–0055
Yang et al., 1993 [41]	rabbit		deletion	–	10.1210/endo.132.5.7682938
Paul et al., 2022 [42]	Xenopus tropicalis	*cyp21a2*	11-base pair deletion	–	10.1210/endocr/bqac182
Eachus et al., 2017 [43]	Zebrafish	*cyp21a2*	c.del211–224, p.P70 fs26X	–	10.1210/en.2017–00549
			c.del212–224, p.P70fs13X	–	

## Data Availability

Not applicable.

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
