# Peer review of "Models of Congenital Adrenal Hyperplasia for Gene Therapies Testing"

_ijms, 2023, doi:10.3390/ijms24065365_

Round 1
Reviewer 1 Report
The review of Glazova et al focuses on models for inherited adrenal gland insufficiency. The review is well structured and the references are appropriate. I have only few comments.
Introduction:
A figure with the metabolic pathway of the synthesis of steroid hormones can be useful.
Pag 2 line 82: "regardless of disease etiology". Please specify if it is referred only to primary adrenal insufficiency
Pag 10 line 420: A paragraph or a table that compare the different models can be useful
Author Response
Dear Reviewer, thank you for the time you have devoted to our manuscript. We have tried to address your comments (see below).
Introduction:
1. A figure with the metabolic pathway of the synthesis of steroid hormones can be useful.
RESPONSE
The general steroidogenic pathways for the discussed models has been added (Fig 1).
2. Pag 2 line 82: "regardless of disease etiology". Please specify if it is referred only to primary adrenal insufficiency
RESPONSE
“The use of cell and gene therapy strategies to create or support functional adrenal tissue is a promising alternative to existing methods of treating adrenal insufficiency, regardless of disease etiology, as this approach allows the organ to function more physiologically.”
was changed to
“The use of cell and gene therapy strategies to create or support functional adrenal tissue is a promising alternative to existing methods of treating primary adrenal insufficiency, regardless of disease etiology, as this approach allows the organ to function more physiologically.”
3. Pag 10 line 420: A paragraph or a table that compare the different models can be useful
RESPONSE
“In the absence of appropriate CAH models, it is difficult to assess the performance of different drug prototypes. Moreover, it is hardly possible at present to assess which of the models is most plausible.“
has been changed to
“In the absence of appropriate CAH models, it is difficult to evaluate the performance of different drug prototypes. Although CAH models are under development (Table 2), it is currently hardly possible to assess which of the models is most plausible.”
Reviewer 2 Report
In the present manuscript, Glazowa et al. have overviewed the available Models of Congenital Adrenal Hyperplasia.
The manuscript contains a section typically for the review article. The introduction and methods are clear and well-written. Please rearrange figure 1 for better understanding for readers (the figure did not contain all necessary information, and all animal models used in the figure need to be described).
The second part of this article is devoted to cell cultures and animal models that need to be improved. Please consider carefully all animal models that might be used to analyze adrenal gland diseases. Also, it would be good to mention in this section about possible limitations and advantages that are commonly shown during working with primary cell cultures.
In the whole article, please check the mentioned gene function. In a few places, the importance of the biological function of described genes is missing (for example, line 73). In the discussion section, literature concerning the analyzed factors is nicely cited.
Moreover, the conclusion section needs to be improved.
Also, the authors must check the entire article and improve it from an editorial angle - remove two spaces, markings, etc.
Author Response
Dear Reviewer, thank you for the time you have devoted to our manuscript. We have tried to address your comments (see below).
1. Please rearrange figure 1 for better understanding for readers (the figure did not contain all necessary information, and all animal models used in the figure need to be described).
RESPONSE
The figure was rearranged.
Figure 1. The main ways to create CAH models. A) Monolayer patient-specific differentiated cell cultures; B) 3D structures from homogenized organ; C) Genome modified animals.
has been changed to
Figure 2. The main types of the CAH models. A) Monolayer patient-specific differentiated cell cultures; B) 3D structures from homogenized organ; C) Genome modified animals (top down): five different mouse models with modified steroidogenic genes, rabbit Cyp11a1 knock-out model, Xenopus tropicalis tadpole and Zebrafish larva cyp21a2 deficiency models.
MSC - mesenchymal stromal cells: SF1 - steroidogenic factor 1; dbcAMP - dibutyryl cyclic adeno-sine monophosphate; ECM - extracellular matrix; KO - knock-out.
Therefore, we added the links to the Figure 2 parts throughout the text to improve usability of the paper.
2. The second part of this article is devoted to cell cultures and animal models that need to be improved. Please consider carefully all animal models that might be used to analyze adrenal gland diseases.
RESPONSE
We added information about more animal models: knocked-out mouse models and rabbit ones. You can kindly find this new information in the article`s Animal models section.
3. Also, it would be good to mention in this section about possible limitations and advantages that are commonly shown during working with primary cell cultures.
RESPONSE
We mentioned that theme into the conclusion section, thank you
4. In the whole article, please check the mentioned gene function. In a few places, the importance of the biological function of described genes is missing (for example, line 73). In the discussion section, literature concerning the analyzed factors is nicely cited.
RESPONSE
“CAH is an example of such a monogenic disorder, caused by mutations in the CYP21A2 gene. ”
was supplemented:
“CAH is an example of such a monogenic disorder, caused by mutations in the CYP21A2 gene. CYP21A2 is an enzyme that is necessary for the synthesis of aldosterone and cortisol (Fig. 1)."
We also added a general steroidogenic pathways with the genes denoted on it. You may kindly find it as a Figure 1.
5. Moreover, the conclusion section needs to be improved.
RESPONSE
We improve the conclusion section with thoughts about the future of the models
6. Also, the authors must check the entire article and improve it from an editorial angle - remove two spaces, markings, etc.
RESPONSE
We have double checked the manuscript to fix all typos.